# Tyrosine Kinase Src Is a Regulatory Factor of Bone Homeostasis

**DOI:** 10.3390/ijms23105508

**Published:** 2022-05-14

**Authors:** Takuma Matsubara, Kazuma Yasuda, Kana Mizuta, Hiroka Kawaue, Shoichiro Kokabu

**Affiliations:** Division of Molecular Signaling and Biochemistry, Department of Health Improvement, Kyushu Dental University, Kitakyushu 803-8580, Japan; r17yasuda@fa.kyu-dent.ac.jp (K.Y.); r18mizuta@fa.kyu-dent.ac.jp (K.M.); r19kawaue@fa.kyu-dent.ac.jp (H.K.); r14kokabu@fa.kyu-dent.ac.jp (S.K.)

**Keywords:** Src, osteoclast, osteoblast

## Abstract

Osteoclasts, which resorb the bone, and osteoblasts, which form the bone, are the key cells regulating bone homeostasis. Osteoporosis and other metabolic bone diseases occur when osteoclast-mediated bone resorption is increased and bone formation by osteoblasts is decreased. Analyses of tyrosine kinase Src-knockout mice revealed that Src is essential for bone resorption by osteoclasts and suppresses bone formation by osteoblasts. Src-knockout mice exhibit osteopetrosis. Therefore, Src is a potential target for osteoporosis therapy. However, Src is ubiquitously expressed in many tissues and is involved in various biological processes, such as cell proliferation, growth, and migration. Thus, it is challenging to develop effective osteoporosis therapies targeting Src. To solve this problem, it is necessary to understand the molecular mechanism of Src function in the bone. Src expression and catalytic activity are maintained at high levels in osteoclasts. The high activity of Src is essential for the attachment of osteoclasts to the bone matrix and to resorb the bone by regulating actin-related molecules. Src also inhibits the activity of Runx2, a master regulator of osteoblast differentiation, suppressing bone formation in osteoblasts. In this paper, we introduce the molecular mechanisms of Src in osteoclasts and osteoblasts to explore its potential for bone metabolic disease therapy.

## 1. Introduction

Bone homeostasis is maintained through remodeling by osteoclasts and osteoblasts [1]. Osteoclasts resorb the bone matrix during the first step of bone remodeling. Subsequently, osteoblasts form new bones by filling in the resorption area [2]. The functional balance between osteoclasts and osteoblasts is important for bone homeostasis. When this balance is disrupted and osteoclast function becomes dominant, bone metabolic diseases such as osteoporosis occur [3,4].

Osteoclasts differentiate from hematopoietic stem cells upon stimulation with macrophage colony-stimulating factor (M-CSF) and receptor activator of nuclear factor kappa-B ligand (RANKL), which are secreted by osteoblasts [5,6]. RANKL binds to the receptor activator of nuclear factor kappa-B (RANK) and activates nuclear factor kappa-B (NF-κB) and mitogen-activated protein kinase (MAPK) signaling [5,7,8,9]. The AP-1 complex consists of c-Jun and c-fos under the MAPK signaling pathway. AP-1 complex and NF-κB then induce nuclear factor of activated T cells 1 (NFATc1), the master regulator of osteoclast differentiation in osteoclast progenitors [5,7,10,11]. Osteoclast progenitors then fuse to form multinuclear cells. These multinuclear cells are mature osteoclasts that attach strongly to the bone matrix by forming a sealing zone [6,12,13]. These attached osteoclasts also form a ruffled border to secrete protons and matrix metalloproteases, such as cathepsin K [6,14,15]. The sealing zone is essential for bone resorption, maintaining the protons and cathepsin K in the bone-resorbing area. 

Osteoblasts are derived from mesenchymal stem cells upon stimulation with growth factors such as bone morphogenetic proteins (BMPs) [2,16]. BMP stimulation induces the expression of the transcription factor Runt-related transcription factor 2 (Runx2); the master regulator of osteoblast differentiation [17]. Runx2 induces the expression of the downstream protein osterix (Osx), an essential transcription factor in osteoblast differentiation [18]. Runx2 and Osx promote osteoblast differentiation in a coordinated and independent manner [19].

The tyrosine kinase Src is one of the molecules that regulate the balance of bone metabolism. Src-deficient osteoclasts have impaired sealing zone formation, ruffled border formation, and subsequent bone resorption activity [12,20]. In contrast to the loss of osteoclast function, the number of osteoblasts is increased and bone formation is accelerated in Src-deficient mice [21]. As a result, Src-deficient mice show osteopetrosis. Although Src is ubiquitously expressed in various tissues and is involved in cell growth, migration, and attachment [22], Src-deficient mice showed no obvious phenotype, except for osteopetrosis. Moreover, treatment with the Src inhibitor dasatinib increased bone mass by suppressing bone resorption and promoting bone anabolism in mouse experiments [23]. These results indicated that Src is a good target for bone metabolic diseases. Src is a proto-oncogene, and Src expression is elevated in high-grade cancers with increased proliferation, invasion, and metastasis. Therefore, Src inhibitors are used as molecularly targeted drugs for some cancers. However, drugs targeting Src are not fully effective for treating osteoporosis. To solve this problem, it is necessary to elucidate the molecular mechanism of Src. Here, we review the molecular mechanisms of Src function in osteoclasts and osteoblasts.

## 2. Src Function in Osteoclasts and Functional Domains

Osteoclasts are differentiated cells, and the number of osteoclasts in Src-deficient mice is greater than that in wild-type mice [12,20]. Thus, Src does not affect osteoclast differentiation but is essential for bone resorption. Src-deficient osteoclasts are unable to resorb bone because they cannot form a sealing zone or ruffled borders to create an acidic environment conducive to osteoclast-mediated bone resorption [20]. The expression and catalytic activity of Src kinases are increased during osteoclast differentiation and remain very high in mature osteoclasts compared to other tissues [24]. Kinase activity is a characteristic function of Src. Schwartzberg et al. demonstrated the importance of Src kinase activity by crossing transgenic mice. In Src-deficient mice, a mutation in lysine 295, the ATP-binding site of Src to activate its kinase activity, to methionine (K295M) did not fully rescue bone resorption activity [25,26,27]. Similar data were obtained from in vitro experiments. Osteoclasts form an actin ring that corresponds to the sealing zone in vitro [28]. Similar to the loss of the sealing zone in vivo, Src-deficient osteoclasts did not form actin rings in vitro. Wild-type Src expression in Src-deficient osteoclasts rescued actin ring formation. In contrast, the K295M mutation did not rescue actin ring formation in Src-deficient osteoclasts [28]. In addition to these data, Src kinase inhibitors such as saracatinib and dasatinib can inhibit bone resorption in vitro and in vivo [3,29]. In summary, Src kinase activity is essential for osteoclastic bone resorption by forming a sealing zone and a ruffled border.

The kinase domain of Src is indeed important, and other regions are also important for Src function. Src consists of four distinct domains: the unique, Src homology (SH) 3, SH2, and kinase domains (Figure 1) [30,31]. The unique domain regulates Src localization. In particular, glycine 2 (G2) of Src is myristoylated. Myristylated Src is localized in lipid rafts, which are sphingosine-rich domains of the cell membrane [30,32,33]. Src localization in lipid rafts is essential for actin organization because actin regulatory proteins assemble in lipid rafts [34]. Src then binds to various proteins through the SH3 and SH2 domains. The SH3 domain is necessary for binding to proline-rich domain-containing proteins, which are target proteins of Src kinase [35]. Replacing tryptophan 118 in the Src SH3 domain with lysine (W118K) does not completely rescue actin ring formation after introduction into Src-deficient osteoclasts [28]. The SH2 domain binds to phosphorylated tyrosine in target proteins [35]. A loss-of-function mutation of the SH2 domain by replacing arginine 175 with leucine (R175L) reduced the actin ring formation ability of Src when overexpressed in Src-deficient osteoclasts [28]. Moreover, an Src-double mutant, W118K and R175L, completely lost its rescue ability [28]. These results indicate that the Src SH3 and SH2 domains are essential for actin ring formation. As described, all four domains of Src are important, but the activity of the kinase domain is the most important; thus, drugs targeting the kinase activity of Src, such as PP2 and dasatinib, have been developed.

## 3. Regulation of Src Activity in Osteoclasts

It is necessary to understand the regulatory mechanism of Src to control Src function in osteoclasts via pharmaceutical interventions. Tyrosine 527 (Y527) at the C-terminus of Src is one of the most important amino acids involved in regulating Src activity. When Src Y527 is phosphorylated, the phosphorylated Y527 binds to its own SH2 domain and hides the kinase active center by changing its conformation [31]. In contrast, a mutant form of Src, wherein tyrosine 527 was changed to phenylalanine (Y527F), constitutively maintained a high kinase activity [36]. Thus, the phosphorylation of Src Y527 is a key process that controls Src kinase activity. Src kinase activity is low in many tissues because Src Y527 is phosphorylated by C-terminal Src kinase (Csk) [27,31]. Interestingly, Csk is ubiquitously expressed, including in osteoclasts; however, there is also high Src activity in osteoclasts [37]. The reason for this is the localization of Src and Csk. Src is predominantly localized in the lipid rafts of the cell membrane through myristylation of the unique domain, regulating actin organization. In contrast, Csk cannot localize to lipid rafts. To regulate Src kinase activity, Csk is recruited to lipid rafts by Csk-binding protein (Cbp) (Figure 2) [31]. Interestingly, Cbp expression is suppressed during osteoclast differentiation [37]. Activated Src is barely suppressed and maintains high kinase activity in osteoclasts. Src is not only suppressed in this context; it is also activated by osteoclast protein tyrosine phosphatase (PTP-oc), a phosphatase that de-phosphorylates the phosphor-Y527 of Src [27,38]. RANKL, M-CSF, and integrin signaling, which promote bone resorption, activate Src through PTP-oc [27]. Since tissues with low Cbp expression are limited, increasing Cbp expression or controlling the localization of Csk would be a potential treatment for bone metabolic disease.

In contrast to the negative regulation of Src activity via Src Y527 phosphorylation, the phosphorylation of tyrosine 416 (Y416) in Src accelerates its kinase activity by stabilizing its active conformation (Figure 1) [39]. Src has a markedly increased kinase activity upon phosphorylation of Y416; thus, the phosphorylation of Src Y416 is a marker of Src kinase activity. The Src inhibitor PP2 negatively regulates the phosphorylation of Src Y416 [40]. PP2 treatment expectedly disturbed osteoclast formation by inhibiting Src kinase activity [41]. Although PP2 has the potential to inhibit Src, its clinical use is only limited to some cancers since Src is also highly activated and involved in metastasis [3,40]. 

The serine 17 (S17) residue in the Src unique domain is phosphorylated by protein kinase A (PKA), activating Ras-related protein 1a (Rap1), which eventually organizes the actin cytoskeleton in neuron-like PC12 cells [42]. However, the exact role of Src S17 remains unclear. Recently, protein phosphatase-1 regulatory unit 18 (PPP1r18) was identified as an Src-binding protein that de-phosphorylates Src S17 by forming a complex with protein phosphatase 1 (PP1) (Figure 2) [43]. PPP1r18 overexpression disturbs actin ring formation in osteoclasts by suppressing Src activity. Thus, Src S17 might be essential for actin ring formation. PPP1r18 expression is considerably reduced during osteoclast differentiation, similar to Cbp levels in osteoclasts and other tissues. In addition, PPP1r18 negatively regulates NFATc1 transactivation by destabilizing c-Fos, which is a complex partner of NFATc1 [9]. NFATc1 and c-fos are upregulated in lymphoid cells during inflammation as they promote the differentiation and function of immune cells, such as T and B cells [44,45]. Therefore, if the expression and function of PPP1r18 are regulated, it would be an effective treatment for osteoporosis, rheumatoid arthritis, and osteoarthritis, which are diseases characterized by immune system-mediated osteoclast function.

## 4. Downstream Proteins of Src in Actin Regulation

Podosomes, in which actin is aggregated into dots, line up around the plasma membrane in a belt-like pattern to form an actin ring [13]. The actin ring is composed of not only actin but another cytoskeleton, tubulin, which also plays an important role. Tubulin extends radially from the center of the cell toward the plasma membrane and arranges podosomes on the plasma membrane in a belt-like pattern. However, Src cannot directly bind to actin and tubulin. To regulate actin and tubulin, Src binds to and phosphorylates various downstream molecules. Here, we introduce these downstream molecules of Src and discuss their potential as therapeutic targets for metabolic bone diseases (Figure 2) [58].

### 4.1. Cortactin

The actin-binding protein cortactin is one of the most major and important proteins in regulating the actin cytoskeleton. Cortactin is involved in actin organization for cell migration and stability. Src phosphorylation of Cortactin allows actin ring formation by regulating actin aggregation [46]. Cortactin is localized in the actin ring and regulates actin aggregation, and tubulin orientation, and determines podosome positioning; osteoclasts deficient in Cortactin do not lose actin rings but have abnormal podosome positioning, resulting in weaker bone resorption activity [47]. Furthermore, the dominant-negative form of Cortactin disturbs actin ring formation and suppresses bone resorption [46]. These results indicate that Cortactin is important for actin ring formation and bone resorption. However, the knockout of Cortactin alone results in only minor reductions in actin ring formation and bone resorption capacity. Additionally, cortactin-deficient mice do not show abnormal bone metabolism [48]. It is possible that hematopoietic-specific protein 1 (HS1), which is highly homologous to Cortactin, compensates for its function. However, it has also been reported that Cortactin and HS1 are not involved in podosome formation in megakaryocytes, and inhibition of Cortactin may have little effect on actin ring formation [59]. Therefore, targeting Cortactin for the treatment of metabolic bone diseases would be problematic. 

### 4.2. p130Cas

p130Cas was identified as a molecule that is hyperphosphorylated in differentiated osteoclasts, but p130Cas is not phosphorylated in Src gene-deficient mice [49]. It is localized in the actin ring, and p130Cas deficiency in osteoclasts results in the failure of actin ring formation and bone resorption. These indicate that p130Cas is phosphorylated by Src and regulates actin ring formation in osteoclasts. p130Cas functions as an adaptor molecule, and in osteoclasts, can bind to not only Src, but also to various other molecules including focal adhesion kinase (FAK), proline-rich tyrosine kinase 2 (Pyk2), and protein-tyrosine phosphatase 1B (PTP1B) [49]. This suggests that p130Cas is not only phosphorylated by Src, but also regulates Src when osteoclasts adhere to the bone matrix. p130Cas does not directly regulate cytoskeletal proteins but forms a complex with Src and Pyk2 to transmit signals to small G proteins such as Rac1 and Rap1, which are involved in actin ring formation [50]. However, this molecule is not a good candidate as a therapeutic target for metabolic bone diseases because p130Cas gene-deficient mice are embryonically lethal.

### 4.3. Proline-Rich Tyrosine Kinase 2 (Pyk2)

Pyk2 is a tyrosine kinase that is highly homologous to FAK and plays an important role in cell migration by regulating the cytoskeleton downstream of integrin signaling. FAK is expressed in various organs and tissues, but Pyk2 is highly expressed in hematopoietic cells, including osteoclasts, and it is localized to the actin ring in osteoclasts. Pyk2 phosphorylation is reduced in Src-deficient osteoclasts. Thus, Pyk2 is considered to function downstream of Src [51]. Pyk2 binds downstream of integrins via the SH3 domain of Src and is phosphorylated. Pyk2 is not only regulated by Src, but when phosphorylated, binds via the SH2 domain of Src and is recruited to the cell adhesion complex containing p130Cas [60,61]. Although less severe than in Src-deficient mice, Pyk2-deficient mice exhibit osteopetrosis due to osteoclast dysfunction [52]. In Pyk2-deficient osteoclasts, tubulin acetylation is suppressed, and this leads to tubulin degradation and the inability to form radial structures. As a result, Pyk2-deficient osteoclasts cannot form actin rings in vitro and are unable to resorb bone [52]. Finally, Pyk2 inhibitors suppress bone loss in ovariectomized mice, a model of osteoporosis, suggesting that Pyk2 is a good therapeutic target for bone metabolic diseases [62].

### 4.4. Casitas B-Lineage Lymphoma Proto-Oncogene (Cbl)

Cbl is an E3 ubiquitin ligase with three homologs: c-Cbl, Cbl-b, and Cbl-c. Although these are highly homologous, they differ in the length of their C-terminal ubiquitin-binding domain (UBA) and are also thought to differ in function [53,61]. In osteoclasts, c-Cbl and Cbl-b have been studied primarily. By comparing phosphorylation levels in Src-deficient and wild-type osteoclasts, Cbl was identified as a molecule that is phosphorylated by Src. c-Cbl phosphorylation during bone resorption is mediated by the binding and mutual activation of Src and Pyk2. In cells where Src activity is maintained at low levels, increased Src kinase activity can ubiquitinate Src and promote Src degradation. In osteoclasts, however, Src kinase activity is maintained at high levels, suggesting the involvement of a different mechanism. c-Cbl is rarely localized to the actin ring in osteoclasts, but is localized with Src in intracellular vesicles. c-Cbl preferentially localizes to the cytoplasm but is translocated to vesicles when phosphorylated by Src. Suppression of c-Cbl expression, in turn, suppresses osteoclast actin ring formation and bone resorption activity in vitro. c-Cbl-deficient mice have minimally reduced osteoclast function, resulting in a slight delay in cartilage to bone replacement [63]. On the other hand, Cbl-b-deficient mice show increased osteoclast function and osteoporosis [64]. This result suggests that Cbl-b may function to regulate the excessive kinase activity of Src. c-Cbl and Cbl-b, despite their high homology, exert opposite effects on bone resorption by osteoclasts. The results from gene-deficient mice suggest that Cbl is not a suitable therapeutic target for bone metabolic disease.

### 4.5. Vav Guanine Nucleotide Exchange Factor 3 (Vav3)

Vav family proteins, the guanine nucleotide exchange factors, activate the Rho family small G proteins. Vav1 is highly expressed in hematopoietic cells, and Vav3 is expressed in various cell types, but especially in osteoclasts. Vav1 and Vav3 are activated by integrins and regulate actin in immune cells. Therefore, Vav1-deficient mice and Vav3-deficient mice were studied. Both mice have increased bone mass, with Vav3-deficient mice the most severely affected [54]. Loss of Vav3 causes a defect in actin ring formation in osteoclasts. Vav3 activates Rac, one of the Rho family proteins, under the M-CSF and integrin signaling. Src also phosphorylates and activates Vav3 by forming a complex with the tyrosine kinase Syk in both M-CSF and integrin signaling. In contrast to Vav3, Vav1 has been reported to act positively and negatively on osteoclasts [65]. No additional studies, such as inhibitors, have been reported for the Vav family, which may make it a difficult target for treatment.

### 4.6. Kinesin Family Member 1C (Kif1c)

Recently, we discovered two novel Src downstream molecules: Kif1c and plectin. Both Src and p130Cas are essential for osteoclast actin ring formation; however, while Src-deficient mice exhibit osteopetrosis, but survive, p130Cas gene deficiency in mice is embryonically lethal. This indicates that Src and p130Cas have common but unique functions. We, therefore, explored the common downstream molecules of Src and p130Cas, via microarray analysis, using mRNA harvested from osteoclasts derived from Src- and p130Cas-deficient mice. This analysis identified Kif1c, a member of the kinesin superfamily that is transported along with tubulin during cargo transport in the cytoplasm [55]. The knockdown of Kif1c using shRNA suppressed actin ring formation, suggesting it is essential for actin ring formation downstream of Src and p130Cas [55]. Kif1c regulates actin accumulation at the cell periphery [66]; therefore, it is implicated in actin ring formation via actin aggregation downstream of Src and p130Cas. These results suggest that Kif1c may be a potential therapeutic target for bone metabolic diseases.

### 4.7. Plectin

We also explored which other molecules bind to Src and regulate the actin cytoskeleton using mass spectrometry and identified the intermediate filament-associated protein plectin [56]. Plectin is a large molecule that binds to not only Src, but also actin, tubulin, intermediate filaments, and other cytoskeletal regulatory proteins [67]. We propose that plectin is essential as a scaffold for Src signaling since the loss of plectin disturbed actin ring formation due to weakened Pyk2 phosphorylation by Src, subsequently inducing tubulin instability [57]. These results suggest that plectin is essential for Src-mediated actin ring formation. A recent report showed that ruthenium complexes inhibited plectin function in cancer [68]. At present, the effects of ruthenium complexes on Src signaling and their function in osteoclasts are unknown; however, the possibility of inhibiting osteoclast function by inhibiting plectin function is an area for the future research. Bisphosphonates, denosumab, and anti-RANKL antibodies, which are widely used to treat osteoporosis, inhibit bone resorption by causing osteoclast loss. However, the loss of osteoclasts causes adverse events, such as microfractures and osteonecrosis of the jaw. Therefore, if we can suppress osteoclast function alone by inhibiting the downstream molecules of Src mentioned above while maintaining their number, we can expect to develop a better therapeutic approach.

## 5. Src in Osteoblasts

Osteopetrosis in Src-knockout mice is caused by decreased osteoclastic bone resorption and increased osteoblastic bone formation [21]. In vitro analysis showed that Src-knockout osteoblasts had decreased alkaline phosphatase (ALP) activity, indicating that Src inhibits osteoblast differentiation [21]. This Src-mediated suppression of osteoblast differentiation, unlike in osteoclasts, is not regulated by actin. It is regulated by the inhibiting Runx2 activity, which upregulates ALP and osteocalcin, markers of osteoblast differentiation. Src promotes the binding of Yes-associated protein (YAP) to Runx2 and negatively regulates Runx2 transcriptional activity in the early stage of osteoblast differentiation [69,70]. Signal transducer and activator of transcription 1 (STAT1) also binds to Runx2, suppressing its nuclear localization, and inhibiting osteoblast differentiation [71]. Src suppresses Runx2 activation by increasing the lifespan of SATA1 [72] (Figure 3). In contrast to its function in osteoclasts, Src kinase activity negatively regulates osteoblast differentiation. Src kinase inhibition by PP2 and dasatinib upregulates osteoblast differentiation markers in vitro [72,73]. 

In contrast to the suppression of Runx2, Src has the potential to promote osteoblast differentiation by activating Osx via BMP signaling [74]. Src binds to and phosphorylates Osx Tyr 64, inducing Osx localization in the nucleus and activating transcriptional activity [74] (Figure 3). This effect of Src is contradictory to that observed in Src-deficient osteoblasts. Osx functions both dependently and independently on Runx2 [18,19]. Osx may have a weaker function in promoting osteoblastic differentiation than Runx2, because ALP activity and mineralization are decreased in Osx-overexpressing cells [19]. Therefore, the phenotype of Src-deficient osteoblasts may be strongly influenced by the derepression of Runx2 activity. In total, Src is considered a drug target because it not only suppresses bone resorption but also promotes bone formation.

There are important points regarding the function of Src in osteoblasts. Src also plays as a sensor for mechanical stress in osteoblasts and osteocytes with actin regulatory proteins such as integrin and p130Cas [75,76]. This mechanical stress is essential for bone formation, as shown by osteoporosis in astronauts in zero gravity and improvement of bone mineral density in osteoporosis patients by mechanical stress [77,78]. Care should be taken in combining treatments that target Src with treatments that cause mechanical stress.

Src is also activated by integrin signaling during cell adhesion or by the inflammatory cytokine tumor necrosis factor (TNF-α) during inflammation via TNF receptor-associated factors (TRAFs), intracellular mediators of TNF-α signaling [79,80]. Src phosphorylates tyrosine near the activation loop of I-kappaB kinase alpha/beta (IKKα/β) for inhibiting NF-κB degradation and facilitating translocation of NF-κB to the nucleus [79]. NF-κB induces the expression of cyclooxygenase-2 (COX-2), which, in turn, induces osteoblast differentiation [79,81]. However, there are reports that NF-κB suppresses osteoblast differentiation [82]. Therefore, the role of Src and NF-κB signaling in osteoblast differentiation requires further investigation.

## 6. Other Src Family Kinases in Bone Metabolism

There are eight members of the Src family of kinases (SFKs): Src, Fyn, Yes, Lyn, Hck, Blk, Lck, and Fgr [31,83]. Although Src is ubiquitously expressed, the phenotype of Src-deficient mice appears only in the bone because SFKs may compensate for Src function in tissues other than the bone. Among these, Fyn, Lyn, and Hck are known to be involved in bone metabolism. Other SFKs are not involved in the functions of osteoclasts and osteoblasts. Single-gene deletions in Fyn do not result in osteopetrosis; however, Fyn and Src double-deletion results in fetal death [84]. Src and Fyn share more than 70% homology in their SH3, SH2, and kinase domains. The expression level of Fyn in osteoclasts is low, and its overexpression in gene-deficient osteoclasts rescues actin ring formation. These results indicate that Fyn has the potential to compensate for Src function but is not expressed at a sufficiently high level in osteoclasts to compensate for the function of Src in Src-deficient osteoclasts (Table 1). Fyn is degraded during differentiation in osteoblasts; however, the function of Fyn in osteoblasts remains to be elucidated (Table 1) [85]. These results suggest that Fyn has a similar function as Src in bone metabolism, but its effect is not strong. Unlike Src, Lyn inhibits osteoclast differentiation and does not affect bone resorption, including the formation of sealing zones, indicating that Lyn is involved in osteoclast formation [86]. Lyn is downregulated during osteoblast differentiation, suggesting that it may suppress osteoblast and osteoclast differentiation (Table 1) [85]. Lyn is considered a factor that inhibits bone turnover. Hence, bone turnover can be controlled if Lyn expression can be regulated. Since fibroblast growth factor 2 (FGF2) downregulates Lyn expression in osteoblasts, FGF2 could also be a target for increasing bone turnover (Table 1) [85]. Hck is an interesting regulator of bone metabolism. Hck single-knockout mice did not show osteopetrosis, but Hck and Src double-knockout mice showed more severe osteopetrosis than Src single-knockout mice (Table 1) [87,88]. Moreover, A-419259, an inhibitor of Hck, suppressed inflammatory bone destruction [89]. However, unlike other members of the Src family of kinases, Hck is upregulated by Runx2 and promotes cell proliferation in chondrocytes that were differentiated from mesenchymal stem cells (Table 1) [90]. Although Hck inhibitors suppressed bone resorption experimentally, as described above, the clinical application of Hck inhibitors is problematic because Hck is also required for bone formation. In our previous study, a chimeric protein replacement of the kinase domain of Hck with that of Src rescued actin ring formation in Src-deficient osteoclasts; however, the Hck wild-type did not have this rescue ability [83]. These results indicate that Hck and Src have overlapping, yet independent roles, in bone resorption. If future studies can reveal the common and different functions of Hck and Src, they will lead to the development of more potent and osteoclast-specific therapies.

## 7. Conclusions

Src is essential for osteoclast-mediated bone resorption through cell adhesion via actin regulation. Its kinase activity is maintained at a high level because of the low expression of Src suppressors in osteoclasts; this kinase activity is essential for regulating the expression of downstream proteins in osteoclasts. Compared to its clear function in osteoclasts, Src has contradictory functions in osteoblasts. However, it has been shown that Src inhibits osteoblast differentiation and suppresses its differentiation. Thus, Src-targeted metabolic bone disease therapy is feasible because it can promote both inhibition of osteoclastic bone resorption and the promotion of osteoblast differentiation. Downstream molecules of Src have been identified. If these can be successfully regulated, it is expected to lead to the discovery and development of better therapies for bone diseases.

## Figures and Tables

**Figure 1 ijms-23-05508-f001:**
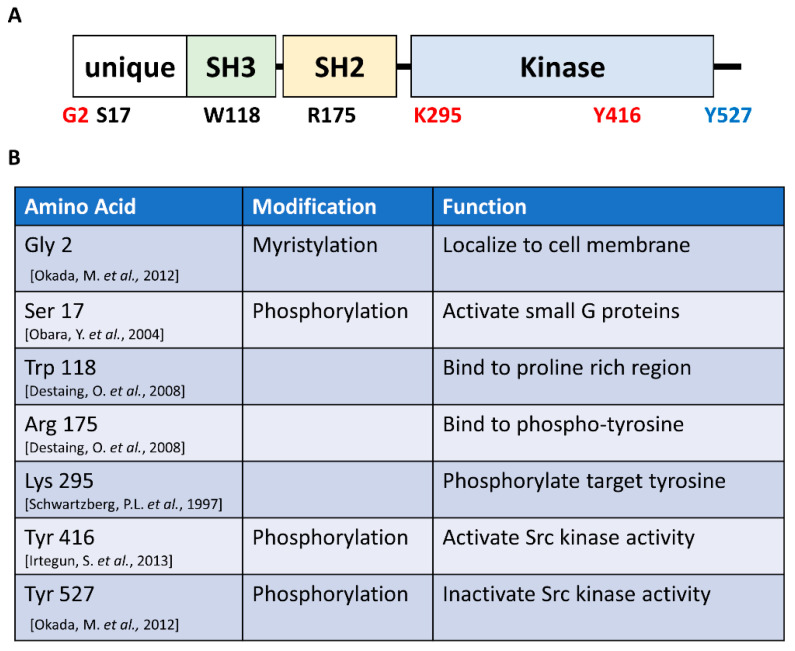
Schema of the domains of Src and the amino acids that play a critical role in its function: (**A**) Src comprises the unique, Src homology (SH) 3, SH2, and catalytic (kinase) domains. (**B**) Functional amino acids in Src.

**Figure 2 ijms-23-05508-f002:**
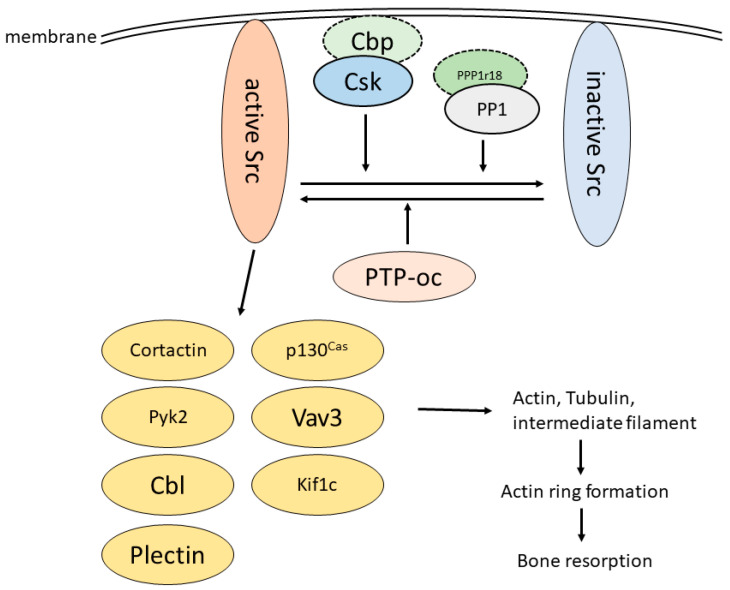
Schema of Src regulation and function in osteoclasts. Src is activated by osteoclast PTP-oc, which de-phosphorylates Src Y527. Src is inactivated through the phosphorylation of Src Y527 induced by the Csk/Cbp complex or the dephosphorylation of Src S16 by the PPP1r18/protein phosphatase 1 (PP1) complex. The expression of these Src-inactivating proteins is low in osteoclasts. Activated Src phosphorylates and activates downstream proteins, such as cortactin [46,47,48], p130Cas [49,50], Pyk2 [51,52], Cbl [53], Vav3 [54], Kif1c [55], and plectin [56,57]. These proteins organize the cytoskeleton to form actin rings in osteoclasts. The circled by solid lines are proteins expressed in osteoclasts. The circled by dotted lines represent those that are less expressed in osteoclasts.

**Figure 3 ijms-23-05508-f003:**
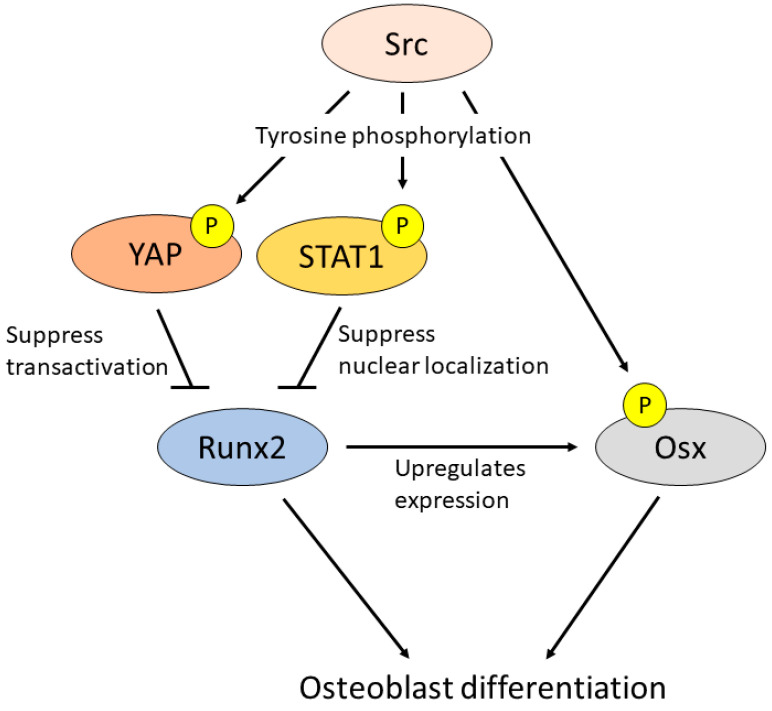
Schema of Src function in osteoblasts. Src activates YAP [69,70] and STAT1 [72] to suppress Runx2 transcription and osteoblast differentiation. In contrast, Src directly binds to and phosphorylates Osx Tyr 64 and induces Osx localization in nuclear to activate transcription to promote osteoblast differentiation [74]. P; phosphor-tyrosine.

**Table 1 ijms-23-05508-t001:** The function of Src family kinases in bone resorption and formation.

Src Family	Bone Resorption	Bone Formation
Src	Essential for osteoclastic bone resorption [12]	Suppress osteoblast differentiation [72,73]
Fyn	Promote osteoclastic bone resorption [83]	Suppress osteoblast differentiation [85]
Lyn	Suppress osteoclast differentiation [86]	Suppress osteoblast differentiation [85]
Hck	Promote osteoclastic bone resorption [87,88]	Upregulate chondrocyte proliferation [90]

## Data Availability

Not applicable.

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
