# Peer review of "Tyrosine Kinase Src Is a Regulatory Factor of Bone Homeostasis"

_ijms, 2022, doi:10.3390/ijms23105508_

Round 1

Reviewer 1 Report

The reviewer mentioned that this manuscript is a well described and comprehensive revision of the regulatory mechanism of tyrosine kinase Src in bone homeostasis, he/she raised concerns which should be addressed.

Add references to functional amino acid in Src in Fig. 1B could help the readers.

Src regulation in osteoblast differentiation, the authors should mention the integrin mediated Src-NfkB signal which leads to COX2 production.

Author Response

Thank you for your kind review and comments. The reviewer's comments are enclosed in [] and my response is written below. Although not mentioned in the comments, we are proofreading the English text again, including the newly added text. The change of the manuscript is shown with “Track Changes” function MS Word.

[Add references to functional amino acid in Src in Fig. 1B could help the readers.]

Thank you for your suggestions. According to your comments, we added references to the functional amino acid of Src in Fig. 1B.,

[Src regulation in osteoblast differentiation, the authors should mention the integrin mediated Src-NfkB signal which leads to COX2 production.]

Thank you for your comment. We have followed your suggestion and added a sentence at the end of chapter 5 about NF-kB signaling regulation by Src at line 334 to 342 (357 to 365 in “view all changes”). Src promotes NF-kB signaling via phosphorylation of IKK at downstream of integrins and TNF, and produces COX2. COX2 promotes osteoblast differentiation, while NF-kB suppresses osteoblast differentiation, so the total effect of Src-NF-kB signaling on osteoblast differentiation requires further study.

Reviewer 2 Report

This is an interesting paper that summarizes the relationship between Src and bone metabolism, and provides a potential target for new osteoporosis drugs. We believe that adding the following information will help the reader understand

  1. Please provide an appropriate citation for line-166-169"Activated Src phosphorylates and activates downstream proteins such as cortactin, p130Cas, Pyk2, Vav3, Cbl, Kif1c, and plectin. These proteins organize the cytoskeleton to form actin rings in 1osteoclasts. " 
  2. Please list the official names of Pyk2, Vav3, Cbl, and Kif1c in line 167-168.
  3. Please specify the meaning of the dotted line in the molecule in Fig. 2.
  4. In Fig3, please provide an explanation of how Src directly induces Osx in the text and cite it appropriately.
  • Please put a half space between the numbers in the quotation.
  • Italicize et al on line 77.

Author Response

Thank you for your kind review and comments. The reviewer's comments are enclosed in [] and my response is written below. Although not mentioned in the comments, we are proofreading the English text again, including the newly added text. The change of the manuscript is shown with “Track Changes” function MS Word.

[Please provide an appropriate citation for line-166-169"Activated Src phosphorylates and activates downstream proteins such as cortactin, p130Cas, Pyk2, Vav3, Cbl, Kif1c, and plectin. These proteins organize the cytoskeleton to form actin rings in 1osteoclasts. "] 

According to your comment, we add the references for p130Cas, Pyk2, Vav3, Cbl, Kif1c and plectin at the figure legend of Fig. 2. Due to the order of references, Fig. 2 was moved from the bottom of Chapter 3 to the bottom of Chapter 4.

[Please list the official names of Pyk2, Vav3, Cbl, and Kif1c in line 167-168.]

We apologize for omit the official names of Pyk2, Vav3, Cbl, and Kif1c.

Because we moved Fig. 2 after Chapter 4 where we described the official names of theses molecules, we did not add the official names at the figure legend.

[Please specify the meaning of the dotted line in the molecule in Fig. 2.]

We apologize for explanatory deficit. We added the explanation of the dotted line that shows less expressed proteins in osteoclasts at figure legend of Figu. 2.

[In Fig3, please provide an explanation of how Src directly induces Osx in the text and cite it appropriately.]

We apologize for explanatory deficit. We added the Src target amino acid in Osx in figure legend and manuscript body at line 319, 320 (340 to 342 in “view all changes”). In addition to this, we revised the figure by adding the functions.

[Please put a half space between the numbers in the quotation.]

We apologize for the mistakes. We put the spaces between numbers in the quotation.

[Italicize et al on line 77.]

We apologize for the mistakes. We revised “et al” on line 75 to italic character.